# Microstimulation of human somatosensory cortex evokes task-dependent, spatially patterned responses in motor cortex

Natalya D. Shelchkova [1,13], John E. Downey [2,13] ✉, Charles M. Greenspon [2,13], Elizaveta V. Okorokova [1], Anton R. Sobinov [2], Ceci Verbaarschot [3,4], Qinpu He[1], Caleb Sponheim [1], Ariana F. Tortolani[1], Dalton D. Moore[1], Matthew T. Kaufman [1,2,5], Ray C. Lee[6], David Satzer[7], Jorge Gonzalez-Martinez[8], Peter C. Warnke[5,7], Lee E. Miller [9], Michael L. Boninger[3,10,11], Robert A. Gaunt [3,10,11,12], Jennifer L. Collinger [3,10,11,12], Nicholas G. Hatsopoulos [1,2,5] & Sliman J. Bensmaia [1,2,5]

The primary motor (M1) and somatosensory (S1) cortices play critical roles in motor control but the signaling between these structures is poorly understood. To fill this gap, we recorded – in three participants in an ongoing human clinical trial (NCT01894802) for people with paralyzed hands – the responses evoked in the hand and arm representations of M1 during intracortical microstimulation (ICMS) in the hand representation of S1. We found that ICMS of S1 activated some M1 neurons at short, fixed latencies consistent with monosynaptic activation. Additionally, most of the ICMS-evoked responses in M1 were more variable in time, suggesting indirect effects of stimulation. The spatial pattern of M1 activation varied systematically: S1 electrodes that elicited percepts in a finger preferentially activated M1 neurons excited during that finger's movement. Moreover, the indirect effects of S1 ICMS on M1 were context dependent, such that the magnitude and even sign relative to baseline varied across tasks. We tested the implications of these effects for brain-control of a virtual hand, in which ICMS conveyed tactile feedback. While ICMS-evoked activation of M1 disrupted decoder performance, this disruption was minimized using biomimetic stimulation, which emphasizes contact transients at the onset and offset of grasp, and reduces sustained stimulation.

Manual interactions with objects involve the integration of sensory signals—about the state of the hand and its interactions with objects—and motor signals—about intended actions. Dexterous hand use relies on both somatosensory and motor cortices as evidenced by the severe deficits in manual dexterity that follow lesions to either of these brain regions[1,2]. However, many of the cortical mechanisms of sensorimotor integration remain to be elucidated. Brodmann's area 1 of somatosensory cortex (S1) has been shown to send projections, albeit sparse

[1]Committee on Computational Neuroscience, University of Chicago, Chicago, IL, USA. [2]Department of Organismal Biology and Anatomy, University of Chicago, Chicago, IL, USA. [3]Rehab Neural Engineering Labs, University of Pittsburgh, Pittsburgh, PA, USA. [4]Psychology and Neuroscience, Maastricht University, Maastricht, Netherlands. [5]Neuroscience Institute, University of Chicago, Chicago, IL, USA. [6]Schwab Rehabilitation Hospital, Chicago, IL, USA. [7]Department of Neurological Surgery, University of Chicago, Chicago, IL, USA. [8]Department of Neurosurgery, University of Pittsburgh, Pittsburgh, PA, USA. [9]Department of Physiology, Northwestern University, Chicago, IL, USA. [10]Department of Physical Medicine and Rehabilitation, University of Pittsburgh, Pittsburgh, PA, USA. [11]Department of Bioengineering, University of Pittsburgh, Pittsburgh, PA, USA. [12]Department of Biomedical Engineering, Carnegie Mellon University, Pittsburgh, PA, USA. [13]These authors contributed equally: Natalya D. Shelchkova, John E. Downey, Charles M. Greenspon. ✉e-mail: johndowney@uchicago.edu

ones, to primary motor cortex (M1)[3–5], and this direct sensorimotor pathway has been hypothesized to play a key role in integrating sensory signals with signals involved in motor execution. Intracortical microstimulation (ICMS) of human S1 has been shown to evoke responses in M1 local field potentials[6,7], and bipolar surface stimulation of monkey S1 evokes responses in M1 neurons[8], both consistent with the identified anatomical pathway. However, the modulation of single-cell responses in M1 to S1 stimulation and the function of the signals passed from S1 to M1 remain to be elucidated.

To fill this gap, we delivered—in three human participants whose hands were paralyzed as a result of a spinal cord injury—ICMS to the hand representation of S1 while we recorded the responses evoked in the hand and arm representation of M1. First, we quantified the prevalence and temporal characteristics of ICMS-evoked activation. Second, we characterized the spatial pattern of activation in M1 and its relationship to the location of the stimulating electrode. Third, we compared ICMS-evoked M1 activity in different task conditions. Finally, we assessed the consequence of the ICMS-evoked activity on our ability to infer on-going motor intent from M1 signals.

## Results

ICMS pulse trains varying in frequency and amplitude were delivered under two conditions: a passive condition in which the participants watched videos and an active condition in which they attempted to reach toward, grasp, and transport a virtual object, a task commonly used to calibrate decoders[9].

### Motor cortex responds to stimulation of somatosensory cortex

First, we examined the responses of M1 neurons to 60-µA, 100-Hz, 1-s ICMS pulse trains delivered through individual electrodes in S1 in the passive condition (see Fig. 1a and Supplementary Fig. 1 for array locations). We found that, ICMS of S1 modulated activity on a majority of M1 channels (Fig. 1b and Fig. 2). For some pairs of M1/S1 channels, the M1 activity increased (Fig. 1b, left), for other pairs, it decreased (Fig. 1b, right). Most modulated M1 channels exhibited both increases and decreases in ICMS-evoked activity, depending on the stimulation channel (48%, 90%, and 98% for C1, P2, and P3, respectively). We verified that these effects were not electrical artifacts by confirming that they were also observed in the responses of sorted single units

(Supplementary Figs. 2 and 3). The prevalence and strength of these effects varied across participants: The effects were stronger and more prevalent in participant C1 than in the other two (P2 and P3, Fig. 2, Supplementary Fig. 4). The participants also differed in the sign of the ICMS-induced modulation, with primarily excitatory responses in C1 (94.2%) and a more even mix in P2 and P3 (39.3% and 46.0% excitatory, respectively).

### Stimulation of somatosensory cortex can directly activate neurons in motor cortex

Next, we examined whether the ICMS-evoked activation of M1 was temporally locked to the stimulation pulses at a short latency, suggesting direct input from S1. To this end, we computed the pulse-triggered average for each pair of stimulating and recording electrodes. We found M1 channels with responses that were systematically locked to the stimulation pulses (Fig. 3a and Supplementary Fig. 5). For most of these channels, the evoked neural activity occurred between 2 and 6 ms after pulse onset with millisecond or even sub-millisecond jitter across pulses (Fig. 3b and Supplementary Fig. 6). To eliminate the possibility that the response latency was longer than the inter-pulse duration, we measured the latency with pulse trains at different frequencies (25, 50, and 100 Hz) and found the latency to be consistent (Supplementary Fig. 7). Of the M1 channels that were modulated by ICMS delivered to S1, 37%, 0.6%, and 32% exhibited this pulse-locked response in C1, P2, and P3, respectively. In contrast to these channels, which seem to receive direct input from S1, most channels exhibited large and significant ICMS-evoked shifts in firing rate with no pronounced peak in the pulse-triggered average (Fig. 3c and Supplementary Fig. 8). Thus, while some of the ICMS-evoked activity in M1 seems to be triggered through direct, possibly monosynaptic, connections from S1, most of it seems to reflect more indirect effects.

### The spatial pattern of activation in motor cortex varies systematically across stimulating electrodes

Next, we examined the spatial patterns of activity evoked over the M1 surface (both direct and indirect) by ICMS in S1 and assessed whether the patterns differed systematically across stimulating electrodes. We found that different stimulating electrodes evoked different spatial patterns of activation in M1 (Supplementary Fig. 9). Moreover, these patterns changed systematically: neighboring stimulating electrodes tended to produce more similar patterns of M1 activation than did distant stimulating electrodes, which sometimes produced entirely non-overlapping patterns. This was particularly pronounced when comparing the spatial pattern of M1 activation evoked by electrodes in different S1 arrays: M1 activation patterns evoked by two electrodes on the same S1 array were significantly more correlated than the patterns

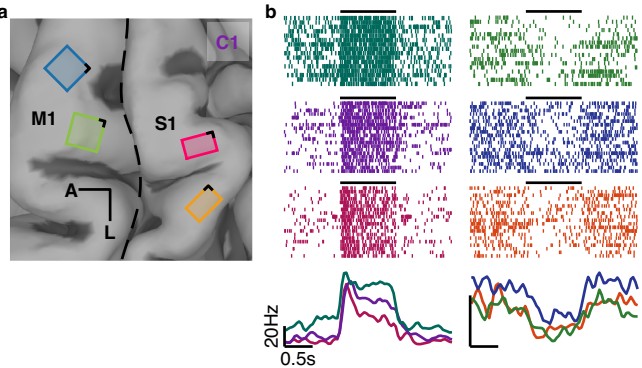

**Fig. 1 | Array placements and interactions. a** Four NeuroPort electrode arrays (Blackrock Neurotech, Inc.) were implanted in the hand and arm representations of motor cortex (M1) and the hand representation of somatosensory cortex (Brodmann's area 1, S1). Here, the implant locations are shown for participant C1. The implant locations for the other two participants are shown in Supplementary Fig. 1. Black lines indicate the posterior-medial corner of each array, which is used as a reference in later figures. **b** M1 responses to ICMS trains delivered to S1. Responses of three example motor channels (spike rasters above and averaged, smoothed firing rates below) that were excited by ICMS (left) and three that were inhibited by ICMS (right). Black horizontal lines indicate the period of ICMS. The green rasters are from participant P3, the rest are from participant C1.

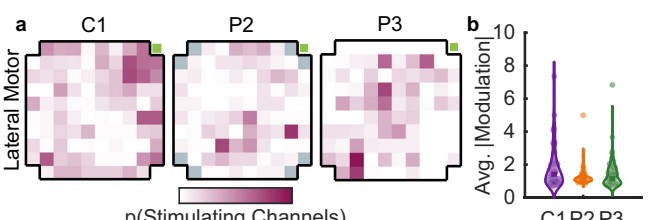

**Fig. 2 | Prevalence of ICMS-evoked activity in motor cortex. a** Proportion of stimulating channels that significantly modulated each motor channel on the lateral motor array of each participant (range: 0–0.7). In P2, gray squares indicate channels that are not wired. The majority of M1 channels could be modulated by ICMS through at least one S1 channel. The green square indicates the posterior-medial corner of the array (see Fig. 1a and Supplementary Fig. 1). **b** ICMS-driven modulation of activity in individual M1 channels, averaged across stimulating channels. Modulation is the ICMS-driven change in the response, normalized by baseline activity.

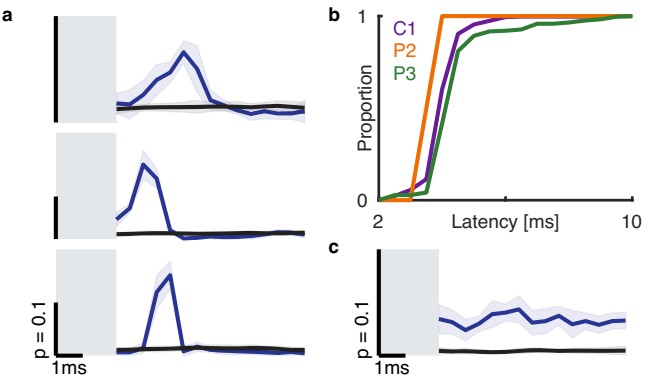

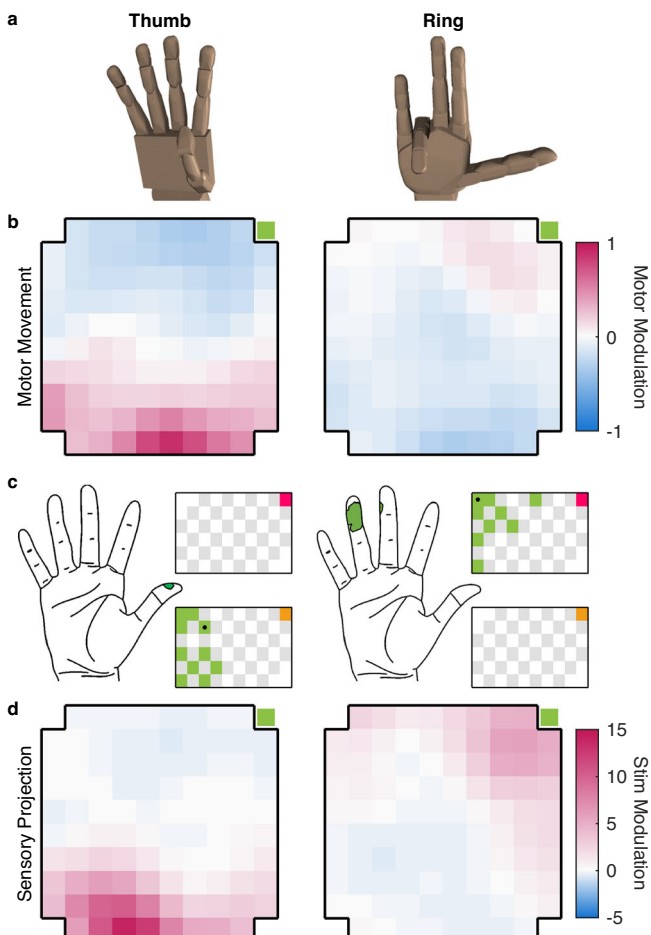

**Fig. 3 | Short-latency, pulse-locked responses in M1. a** Pulse-triggered average of the responses of three motor channels to ICMS at 100 Hz. On a subset of channels, such as these, responses were tightly locked to each pulse with millisecond or even sub-millisecond jitter across pulses. Blue line denotes the response during stimulation, black line the response during baseline (sham stimulation), gray box indicates blanked recording time to eliminate the stimulation artifact. Error bars represent bootstrapped standard error. Scale bar indicates a 10% probability of a spike occurring in a 0.5-ms bin. **b** Cumulative distribution of the latency of the peak pulse-locked, direct, response. Latencies tended to be shorter than 6 ms. **c** Pulse-triggered average of the response of a motor channel whose activity increases with stimulation but is not pulse locked. Error bars represent bootstrapped standard error.

evoked by two electrodes on different arrays ($p < 0.001$, Wilcoxon Rank-Sum test for each of the three participants, Supplementary Fig. 9). Within array, the correlation decreased as the distance between the two S1 channels increased (Supplementary Fig. 10). While the trends were similar, P2 showed lower correlations in ICMS-evoked spatial patterns across all M1/S1 pairs.

Examination of the spatial patterns of M1 activation suggested a coordinated progression of effects across the S1 arrays. In participant C1, for example, lateral stimulating electrodes tended to activate neurons on the lateral aspect of the M1 array and medial stimulating electrodes tended to activate more medially located M1 neurons (Supplementary Fig. 9A). We hypothesized that this progression reflects the respective somatotopic organizations of S1 and M1. For example, stimulation through electrodes in the S1 thumb representation might preferentially activate neurons in the M1 thumb representation. To test this hypothesis, we mapped the somatotopic organization of M1 by measuring, on each motor channel, the evoked activity when the participant attempted to move each digit. For each motor channel, we computed the difference between the activation evoked during attempted movement of each digit and the mean activation during movement of each of the five digits (motor map, Fig. 4a, b). This analysis gauged the extent to which a motor channel responded more during attempted movement of some digits than others. We mapped the somatotopic organization of S1 by identifying the digit on which the participant reported the sensation when stimulation was delivered through each electrode (the projected field, PF, Fig. 4c). Having constructed these motor and sensory maps, we then derived the pattern of M1 activation when ICMS was delivered through electrodes with PFs on each digit in turn (sensory projection map, Fig. 4d). Finally, we assessed the degree to which the motor map matched the sensory projection map. To this end, we compared the activation evoked in individual M1 channels by stimulation through somatotopically matched S1 electrodes, that is those with PFs on the digit that maximally activated the M1 channel during attempted movement, to the activation evoked by stimulation through unmatched electrodes. We found the M1 activation was greater for somatotopically matched than unmatched pairs ($p < 0.001$, Wilcoxon Rank-Sum test) for participants C1 and P3 (Fig. 5a, Supplementary Fig. 11).

**Fig. 4 | Shared somatotopy between movement-evoked and ICMS-evoked activity in participant C1. a** Rendering of the extrema of thumb and ring flexion in virtual reality. **b** Z-scored difference in firing rate during attempted flexion of the thumb (left) and ring finger (right) vs. the mean activation during attempted flexion of each of the 5 digits. The green square indicates the posterior-medial corner of the array (Fig. 1a). **c** The green regions on the hand diagrams denote the projected fields reported by participant C1 when stimulated through one channel in the lateral and medial sensory array, respectively (indicated by a black dot in the array maps). Channels shaded in green on the array diagram denote electrodes with projected fields on the thumb and ring finger, respectively. Channels shaded in gray denote unwired electrodes. Pink and orange squares in the top right indicate the posterior and medial corner of the medial and lateral sensory array, respectively (Fig. 1a). **d** Average M1 activity evoked by stimulation through S1 channels with projected fields on the thumb (left) and the ring finger (right). Motor channels that respond strongly to attempted thumb or ring finger movements tend to also be strongly activated by stimulation of electrodes with projected fields on the thumb or ring finger, respectively. Green squares indicate the posterior and medial corner of the array (Fig. 1a).

As most M1 channels responded to multiple digits and could be activated even by unmatched S1 channels, albeit to differing degrees, we next assessed whether the full pattern of M1 activity evoked during attempted single-digit movements was predictive of the digit dependence of the ICMS-evoked activity. For example, would an M1 channel that responded most to attempted movement of the thumb, then index, then middle finger be most susceptible to stimulation through S1 channels with thumb PFs, less susceptible to stimulation through S1 channels with index PFs, and least susceptible to channels with middle finger PFs? For participants C1 and P3, we observed the hypothesized result across M1 channels (Fig. 5b, Kruskal–Wallis $p < 0.001$) and in single M1 channels (Supplementary Fig. 12). These results are consistent with the hypothesis that electrical activation of S1 neurons leads

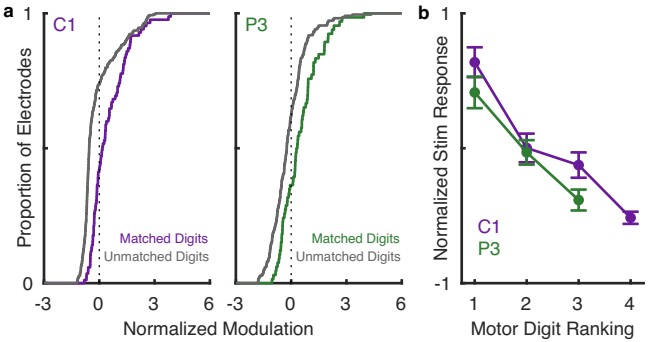

**Fig. 5 | M1 is somatotopically linked to S1. a** In participants C1 and P3, M1 electrodes are more susceptible to ICMS delivered through S1 electrodes whose projected fields match the movement fields. **b** When the dominant movement field matches the dominant digit in the projected field, the susceptibility is strongest; when the second most dominant movement field matches the dominant digit projected field, the susceptibility is weaker; etc. Lines denote the mean, error bars the standard error of the mean, *n* = 96 channels.

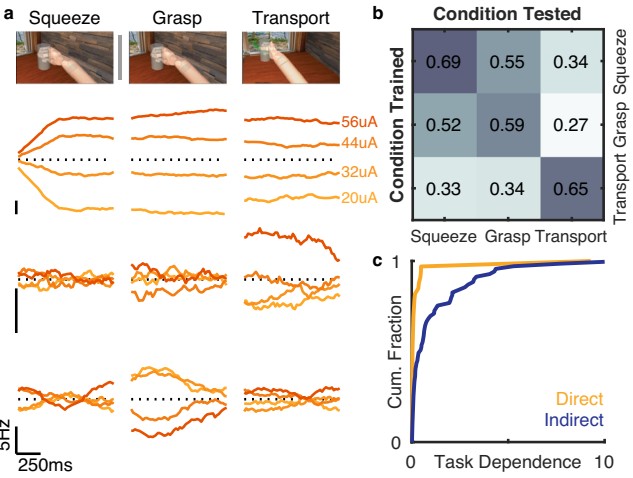

**Fig. 6 | ICMS-evoked activity depends on behavior. a** Top: Squeeze, grasp, and transport in the VR environment. Bottom: Three example motor channels from Participant C1 exhibit different responses to four levels of ICMS across three motor conditions (squeeze, grasp, transport). Traces denote the firing rate evoked by stimulation at the four levels after subtraction of the mean across conditions. **b** Stimulation amplitude classifier performance. Classifiers were trained from M1 activity on one of the three conditions and tested on activity in each condition (cross-validated within condition). **c** Task dependence–gauged by the strength of the condition/amplitude interaction divided by the strength of the main effect of amplitude–is nearly zero for the pulse-locked responses (direct) but varies widely for the non-pulse locked (indirect) ones. The units with direct input from somatosensory cortex respond the same way to ICMS across behavioral conditions.

to preferential activation of M1 neurons with matching movement fields in participants C1 and P3.

The somatotopic link between S1 and M1 was much weaker and, in fact, non-significant in participant P2. Note, however, that ICMS-driven M1 activation in this participant was sparse (Fig. 2a), weak (Fig. 2b), and unpatterned (Supplementary Fig. 9C, D). We attribute the lack of spatial patterning to the fact that this participant's most lateral M1 array, more medial than its counterparts in the other two participants, was located in the proximal limb representation as evidenced by robust arm- but weak digit-related activity. The M1 arrays in participant P2 are also much older than are those in participants C1 and P3, which may have contributed to the observed differences, though robust movement-related signals could still be harnessed from this array (Supplementary Fig. 13).

**Stimulation-evoked activation in motor cortex differs across tasks**

The analyses shown above were carried out on M1 responses collected when the participants were not engaged in any motor task. Because manual touch typically occurs in the context of active interactions with objects, we next examined whether the signaling between S1 and M1 might depend on motor behavior. To this end, we measured ICMS-evoked responses in M1 as participants C1 and P3 performed two tasks. In the first task (squeeze), they attempted to squeeze a cylinder in a virtual reality environment (i.e., without making any overt movement). In this task, contact with the virtual cylinder triggered ICMS (frequency = 100 Hz) through two electrodes delivered at one of four amplitudes (20, 32, 44, and 56 μA), presented in random order. The participants were instructed to report the magnitude of the percept evoked by the ICMS train to ensure their engagement. In the second task (grasp and transport), the participants observed and attempted to mimic the actions of a virtual limb as it reached for and grasped the cylinder in one location and transported it to a different location. Upon grasp, the same ICMS trains were delivered as in the squeeze task (again ordered randomly across trials) until the object reached the target location. For this task, we analyzed the responses during the grasp phase and the transport phase separately. We reasoned that the grasp epoch involved the same behavior as did the squeeze task, whereas the transport phase involved a different behavior. We then compared M1 responses to ICMS across the three conditions ("squeeze," "grasp," and "transport").

We first verified that M1 was engaged in the two behavioral tasks by examining the task dependence of the M1 activity. We found that activity on most motor channels differed across task conditions

(squeeze vs. grasp vs. transport, >80% of the electrodes exhibited significant task modulation according to a multi-way ANOVA, *p* < 0.05 in both participants). Moreover, the observed reach endpoint could be decoded during the grasp and transport task from the M1 population activity (84 and 87% classification accuracy for two sessions with participant C1 and 26% accuracy for participant P3; chance = 12.5%; in participant P3, the motor arrays were much more strongly modulated by hand/wrist than shoulder movements, thus the poor performance).

Examining the dependence of the M1 activity on ICMS amplitude, we found that many motor channels were modulated in an amplitude dependent way, generally exhibiting higher firing rates at higher ICMS amplitudes (across participants and pairs of stimulating electrodes, 31% and 78% for two sessions with participant C1 and 54% for one session with participant P3; *p* < 0.05 multi-way ANOVA, Supplementary Fig. 14). Surprisingly, however, the effect of ICMS varied across tasks: the responses of some M1 neurons were strongly modulated by ICMS during some tasks but not others. Even the squeeze and grasp conditions sometimes yielded different ICMS-evoked M1 activations, even though the behavior is nearly identical–the only difference being that grasp occurs at the end of a reach and just before transport whereas squeeze is a single, isolated movement. To quantify the task dependence, we computed the interaction between task and ICMS amplitude and found that a large number of M1 units yielded a significant interaction (17% and 42% for the two sessions with participant C1 and 19% for the session with participant P3, *p* < 0.05) (Fig. 6a, Supplementary Fig. 15A).

To further demonstrate the task dependence of the ICMS effects, we built a classifier of ICMS amplitude based on responses obtained in one of the three conditions (squeeze, grasp, transport) and attempted to use it to decode ICMS amplitude from the responses in the other two conditions. We found that, while we could decode ICMS amplitude on held-out data within condition with up to 69% accuracy in C1, performance was worse across conditions (Fig. 6b, Supplementary Fig. 15B). In particular, the effects of ICMS during transport were very

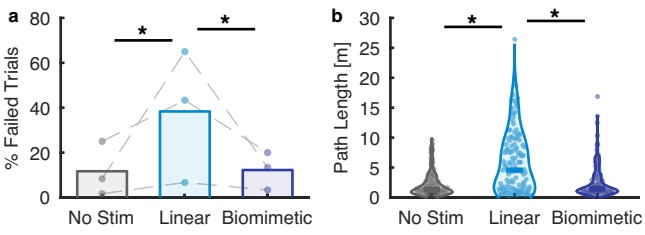

**Fig. 7 | Decoder performance with and without sensory feedback (from participant C1). a** Failure rate for the three conditions. Rates collected during a single session are connected by a dotted line. **b** Path length during the transport phase with different stimulation conditions. Linear stimulation caused the path length of the transport phase to be significantly longer than without stimulation ($p = 10^{-12}$, K-S Test, two-sided). In contrast, biomimetic stimulation was significantly more efficient than its linear counterpart ($p = 10^{-10}$) and not significantly different from the no-stimulation condition ($p = 0.4$).

different from during squeeze or grasp as evidenced by the poorer performance of classifiers built on the former and tested on the latter (33% accuracy).

The task dependence of ICMS-evoked M1 activity varied idiosyncratically across M1 channels. For example, one M1 channel was susceptible during the transport phase but not the grasp phase while another channel showed the reverse pattern of responses (Fig. 6a). This heterogeneity suggests that the task-dependence of the modulation was not driven by array-wide differences in the baseline activity across conditions, for example, reflecting saturation due to higher baseline activation in some conditions. Even at the single channel level, the modulation strength was not systematically related to the (task-dependent) baseline firing rate ($p > 0.05$ for both participants, Friedman's Test, Supplementary Fig. 16). This heterogeneous pattern of task-dependent susceptibility to ICMS implies that ICMS-evoked activity in M1 cannot be straightforwardly distinguished from intrinsic motor-related activity.

Interestingly, the task dependence of the susceptibility to ICMS was not observed for M1/S1 pairs that exhibited pulse-locked responses (Fig. 6c), suggesting that this dependence does not reflect a change in the direct input from S1 but rather a change in the impact of this input on M1.

### ICMS-evoked M1 activity contaminates motor decoding
In BCIs, signals from M1 are often used to infer motor intent and control the bionic limb[10]. The observed contamination of these intrinsic signals about intended movement with contact-related signals stemming from ICMS to S1 is thus liable to interfere with motor decoding and degrade the function of the bionic limb. To investigate this possibility, we trained a decoder (Optimal Linear Estimator)[9] to control a virtual arm across three translational degrees of freedom to enable C1 to reach to an object, grasp it, and transport it to a new location. The grasp was automatically triggered once the hand reached the object's location, to decouple the ICMS from the grasp kinematics, thereby ensuring that the ICMS was identical across grasps. During object contact, ICMS at 100 Hz and 52 µA was delivered through two electrodes (with PFs on the thumb and index), evoking a sensation whose strength was commensurate with the grasp force required to maintain object grasp (i.e., sufficient to evoke a moderately strong tactile sensation). The stimulation led to significantly more trials in which the participant was unable to complete the transport within the allotted 10-s window (38% and 12%, with and without stimulation, respectively, $p < 0.001$, chi-squared test; Fig. 7a). These failures were primarily due to increases in path length—the distance traveled to reach the target—during transport with stimulation compared to without (4.6 m vs. 1.4 m, $p < 0.001$, two-sample Kolmogorov–Smirnov test; Fig. 7b). In other words, the ICMS contaminated the M1 activity

used by the decoder to infer on-going motor intent. The disruption is likely to be far stronger when decoding hand (rather than arm) movements given that ICMS-driven activity in M1 is strongest for somatotopically linked segments.

### Biomimetic somatosensory feedback rescues decoder performance
Importantly, because ICMS-evoked activity in M1 is task dependent, its influence on a decoder cannot be easily predicted and eliminated. However, we reasoned that reducing the intensity of ICMS would reduce its deleterious effects. With peripheral nerve interfaces, biomimetic somatosensory feedback—characterized by high-amplitude phasic stimulation at the onset and offset of contact, and far weaker stimulation during maintained contact[11,12]—has been shown to elicit more natural and intuitive sensations[13,14]. In the present context, we reasoned that this feedback might offer the additional benefit of reducing the total amount of stimulation and thus be less deleterious to decoding. To test this possibility, we had the participant perform the reach, grasp, and transport task but provided information about grasp force using biomimetic ICMS-based feedback. With biomimetic feedback, the onset and offset transient amplitudes were higher than the highest amplitude used in the standard trains—termed "linear" because they track the applied force – but sustained stimulation was weaker (32 vs. 52 µA). With this feedback, the participant transported objects with a third as many failures compared with linear stimulation (12% vs. 38%, $p < 0.001$, chi-squared test; Fig. 7a). This improved performance was characterized by shorter movements during the transport phase (1.4 m vs. 4.6 m $p < 0.001$, two-sample Kolmogorov–Smirnov test; Fig. 7b). In fact, performance with the biomimetic feedback was nearly identical to that with no stimulation (12% vs. 12% failure rate, $p = 0.87$, chi-squared test; mean path length was numerically identical for both conditions at 1.4 m; Fig. 7). Note that ICMS-feedback did not have any beneficial effects on performance (as it has been previously shown to do[15]) because grasping was automated and therefore did not require or allow for any online correction.

## Discussion
We show that ICMS of S1 evokes widespread activity in M1. Some of this activity takes the form of short-latency responses to ICMS that are time-locked to individual ICMS pulses. Most of the ICMS-driven activation in M1 is not pulse-locked, however, and seems to reflect an indirect effect of S1 input. In both cases, the spatial pattern of evoked activity in M1 depends systematically on the location of the S1 stimulating electrode: an M1 channel is susceptible to being modulated by an S1 channel to the extent that they both encode a matching part of the hand. The signals that are directly transmitted from S1 to M1 are consistent across tasks, but their indirect impact on M1 activity is highly task dependent and varies widely across pairs of S1/M1 channels. Finally, ICMS-evoked M1 activity is relevant to prosthetics as it disrupts the ability to decode motor intent. However, this disruption can be minimized with a more biomimetic form of somatosensory feedback, which emphasizes the transient phases of object contact (its onset and offset) and minimizes sustained ICMS, mirroring the patterns of neuronal activity during object interactions.

In both humans and macaques, Brodmann's area 1 and M1 have been shown to be connected anatomically[3,4]. In macaques, tracer injections in area 1 reveal reciprocal connections with M1[3,5], albeit sparse ones. In humans, probabilistic diffusion tractography reveals strong connections between area 1 and M1[4]. Microstimulation of human somatosensory cortex with either surface or penetrating electrodes has been shown to evoke field potentials in motor cortex[6,7], revealing a functional correlate to the anatomical findings. However, neither the time course of these signals nor their spatial specificity could be gleaned from these measurements of aggregate neuronal activity in M1. While short latency ICMS-evoked responses have been

found across sensorimotor cortex in other organisms[8,16–20], the present report is the first to document systematic signaling between somatosensory and motor cortices of humans at the cellular level. In macaques, surface stimulation of S1 was shown to evoke responses in M1 with latencies ranging from ~1 to 7 ms[8], consistent with our results. Because we discard the first 2 ms of the response after pulse onset to avoid contamination from the stimulation artifact, we likely missed some responses that occurred at shorter latencies. Some of the short-latency, low-jitter M1 responses to ICMS in S1 may reflect antidromic activation[21,22], but the latency, jitter, and spiking probabilities of the pulse-locked responses were smoothly distributed over a range, offering no hint of a separation between two classes of activation (antidromic vs. orthodromic) (Supplementary Fig. 6).

We found that the functional connectivity between S1 and M1 is patterned: neighboring electrodes in S1 produce similar spatial patterns of activation in M1. Moreover, this patterning follows somatotopic maps in S1 and M1: a given channel in S1 is liable to activate a given channel in M1 to the extent that these encode overlapping parts of the hand. The somatotopic patterning in M1 seems at odds with the observations that individual M1 neurons encode movements of joints distributed over the entire hand[23–25], resulting in a coarse somatotopic organization. Nonetheless, we observed a somatotopic progression over the sampled cortex, even within the M1 hand representation. The pattern of digit preferences in the movement fields of an individual M1 channel mirrored the pattern of digit preferences in the S1 channels that were most effective in activating that channel. The somatotopic organization of the S1-M1 connectivity is consistent with the interpretation that sensory feedback from a given digit preferentially informs the ongoing motor control of that digit. Note, however, that we were also able to decode reaching movements from the putative hand representation in M1, arguing this somatotopic organization is not absolute, consistent with prior findings[26]. The somatotopically linked connectivity observed here accounts for the observation in macaques that M1 neurons receive tactile input on the associated hand segment[27], the provenance of which had not been established. The anatomical substrates for this somatotopically linked connectivity have been previously established at the level of arm vs. hand[5,28], but our observation at the level of fingers suggests an even more specific somatotopic link. Analysis of the nature of these signals during natural manual interactions in intact humans and monkeys will shed further light on the functional role of this cortico-cortical signaling.

ICMS of hand S1 has been shown to elicit vivid sensations that are experienced on the hand[10,29–32]. These sensations can be used to provide tactile feedback about object interactions and have been shown to improve the functionality of a brain-controlled robotic hand[15]. In the one demonstration of the benefits of somatosensory feedback on object manipulation, however, the participant's motor arrays, which were located in the proximal limb representation of M1, were only weakly impacted by ICMS to hand S1 (P2 in this study, see Supplementary Fig. 1). When M1 and S1 arrays are both in the respective hand representations, as in C1 and P3, ICMS has a deleterious effect on decoding, thereby counteracting—at least in part—any benefits of sensation. The fact that the majority of ICMS-induced activity in M1 is dependent on behavior implies that mitigating the impact of ICMS on decoding will be challenging. Indeed, training a decoder based on combined observation and stimulation will work only (1) if the decoder is trained on tasks that span the space of possible behaviors and (2) if the subspace of ICMS-evoked activity in M1 is largely non-overlapping with that involved in motor control[33]. The first condition will be difficult to meet given realistic time constraints, and we have evidence that the second condition is not met (Supplementary Fig. 17). Fortunately, we were able to eliminate the impact of ICMS on decoding by implementing phasic biomimetic feedback, designed to mimic natural cutaneous responses in cortex. Indeed, throughout the somatosensory

neuraxis, neural populations respond more strongly at the onset and offset of object contact and much more weakly to maintained contact[34,35]. Sensory feedback with this property entails weaker ICMS during maintained grasp, thereby resulting in weaker and thus less disruptive effects of ICMS in M1 (Supplementary Fig. 12). In studies on electrical interfaces with the peripheral nerve, biomimetic sensory feedback has been shown to be more intuitive and naturalistic[13,14,36]. In studies with BCIs, we have shown that biomimetic ICMS yields more precise force feedback[37]. Here, we show that biomimetic stimulation also alleviates the disruptive effect of ICMS on decoding performance for brain-controlled bionic hands.

ICMS in S1 reveals strong signaling from S1 to M1 that is patterned such that S1 neurons with projected fields on one hand region preferentially activate M1 neurons that are implicated in moving that part of the hand. While the (seemingly) direct connection between S1 and M1 is fixed, the overall impact of ICMS to S1 on M1 activity is task dependent. This channel of communication between S1 and M1 disrupts the decoding of motor intent from M1 signals, but this disruption can be minimized using biomimetic feedback.

## Methods

This study was approved by the institutional review boards at the University of Chicago and the University of Pittsburgh (Pittsburgh, PA, USA), and was carried out under an investigation device exemption (IDE) from the FDA.

### Participants

The three participants, part of a multi-site clinical trial (registered on clinicaltrials.gov, NCT01894802), provided informed consent including prior to any experimental procedures. The primary eligibility criterion for the clinical trial was paralysis of at least one hand following spinal cord injury or brain-stem stroke. The primary exclusion criteria were any health concerns that were likely to be exacerbated by surgery or brain stimulation (e.g., chronic pressure sores or a history of seizures). All three participants were male between the ages of 28 and 57 at time of implant and presented with SCI that occurred between 10 and 35 years prior. The primary outcome of the ongoing trial is that the implant is safe for at least 1 year; all enrolled participants have exceeded this goal. The secondary outcome was functional use of the device; assessment of this outcome is still active. The results presented here do not contribute to the assessment of these outcomes. Participant C1 presented with a C4-level ASIA D spinal cord injury (SCI). He had no spared control of the intrinsic or extrinsic muscles of the right hand but retained the ability to move his arm with noted weakness in many upper limb muscles. Filament tests revealed spared deep sensation but diminished light touch in the right hand (detection thresholds ranged from 0.6 to 2.0 g across digit tips). Data were collected 1–1.5 years after implant. Participant P2 presented with a C5 motor/C6 sensory ASIA B SCI. He had no spared control of the intrinsic or extrinsic muscles of the right hand but had limited control of wrist flexion and extension. Proximal limb control at the shoulder was intact, as was elbow flexion. However, he had no voluntary control of elbow extension. He was insensate in the ulnar region of the hand (digits 3–5) on both the palmar and volar surfaces but retained both diminished light touch and deep sensation on the radial side (digits 1–2) (thresholds were 1.4 g to 8 g on the thumb and index, respectively, and 180 g on the middle finger). Data were collected 6.5 years after implant. Participant P3 presented with a C6 ASIA B SCI. He had no functional control of the intrinsic or extrinsic muscles of the right hand but retained the ability to move his arm with noted weakness in many upper limb muscles. He was insensate in the ulnar region of the hand on both the palmar and volar surfaces but retained diminished light touch and deep sensation on the radial side (thresholds were 0.07 g and 1.6 g on the thumb and index and 8 g on the middle finger). Data were collected 1.5–2 years after implant. All participants were

compensated for time spent on the study, receiving $1080 per month that the devices were implanted for testing.

## Statistics & reproducibility

This project was part of an ongoing clinical trial where blinding was not possible. All enrolled subjects at the time of data collection were included, but there was no statistical method used to predetermine sample size. No data were excluded from analyses.

## Array implantation

We implanted four NeuroPort microelectrode arrays (Blackrock Neurotech, Salt Lake City, UT, USA) in the left hemisphere of each participant. Two of the arrays, implanted in somatosensory cortex (Brodmann's area 1, S1), were 2.4 × 4 mm, each with sixty 1.5-mm electrode shanks wired in a checkerboard pattern such that 32 electrodes could be stimulated. The other two arrays, implanted in motor cortex (M1), were 4 × 4 mm with one hundred 1.5-mm electrode shanks, 96 (participants C1 and P3) or 88 (participant P2) of which were wired (active). Four inactive shanks were located at the corners of all arrays (with an additional 8 for participant P2). In P2, the motor cortex arrays were metalized with platinum while the somatosensory arrays with coated in sputtered iridium oxide. In participants C1 and P3, all electrodes were coated with sputtered iridium oxide. Most of the electrodes (74/96) on the medial array of participant C1 were too noisy to yield useful data and deactivated. Each participant had two percutaneous connectors placed on their skull, with each connected to one sensory and one motor array. We targeted array placement during surgery using functional neuroimaging of the participants attempting to make movements of the hand and arm, and imagining feeling sensations on their fingertips[29], within the constraints of anatomical features such as blood vessels and cortical topography (Fig. 1a and Supplementary Fig. 1). Array locations, shown in Fig. 1a and Supplementary Fig. 1 on structural MRI models of each participant's brain, were confirmed using intraoperative photographs after insertion.

## Neural stimulation

Stimulation was delivered using a CereStim microstimulator (Blackrock Neurotech, Salt Lake City, UT, USA). Stimulation pulses were cathodal first, current controlled, and charge balanced, over a range that has been previously deemed safe[38]. Each pulse consisted of a 200-µs long cathodal phase, then a 100-µs interphase period followed by a 400-µs long anodal phase at half the cathodal amplitude. Stimulation pulses could be presented at up to 300 Hz. Further details on selection of stimulation parameters can be found in ref. 29.

## Neural recordings

Neural signals in M1 were recorded at 30 kHz using the NeuroPort system (Blackrock Neurotech, Salt Lake City, UT, USA). Each stimulation pulse triggered a 1.6 ms sample-and-hold circuit in the preamplifier (hardware blanking) to avoid saturating the amplifiers and to minimize transient-induced ringing in the filtered data. The data were high-pass filtered with a 1st order 750-Hz filter[39]. Whenever the signal crossed a threshold (−4.5 RMS, set at the start of each recording session), a spiking event was recorded and a snippet of the waveform was saved. Spikes were binned in 20-ms bins for decoding. To confirm that the observed effects reflect neural activity and not an electrical artifact, we sorted units offline using Plexon Offline Sorter and repeated many of the analyses described below on isolated single units.

## Stimulation protocol—passive condition

To study the effects of stimulation on M1 activity, we stimulated through each S1 channel a minimum of 15 times at 60 µA and 100 Hz in 1 s trains. Electrode order within each array was shuffled and stimulation was interleaved across arrays. The interval between pulse trains was 3 s in participant C1 and a random duration between 3 to 4 s in participants P2 and P3 to counteract any anticipatory effects.

## Gauging the strength of ICMS-driven activity in motor cortex

To understand the effects of ICMS in S1 on activity in M1, we compared the fluctuations in firing during baseline to those during the stimulation interval. For each motor channel, we sampled the difference in firing rate between two consecutive 1-s intervals during the intertrial periods, computed the mean, and repeated this process 1000 times to generate a null distribution of baseline fluctuations over the course of a recording session. For each stimulating channel, we calculated the change in firing rate between a 1 s interval preceding the stimulation train and the firing rate during the stimulation train itself, which gauged the effect of stimulation on each motor channel. For these analyses, we discarded the first 2 ms of the response after each pulse to eliminate any potential electrical artifacts that extended beyond the initial 1.6-ms hardware blanking window. We simulated this blanking in the baseline response to generate the null distribution. Motor channels were considered to be modulated by stimulation if their average change in firing rate during stimulation was significantly different from the null distribution ($p < 0.001$). To gauge the sign and magnitude of the effect of stimulation on a motor channel, we expressed the change in firing rate during stimulation for each motor/stimulation channel pair as a $z$-score based on the null distribution for that motor channel. Positive modulation values indicate an excitatory effect while negative modulation values indicate an inhibitory effect.

## Gauging the timing of ICMS-driven activity in motor cortex

To determine if motor units were phase locked to the stimulation pulses, we computed the pulse triggered average (PTA). Specifically, we binned the spikes evoked during each inter-pulse interval into 0.5-ms bins and computed the probability of spiking in each bin (i.e., the proportion of times a pulse evoked a spike in that bin). To assess whether there was a significant peak in the PTA, indicating a pulse-locked response, we first identified the time at which the probability of a spike occurring was highest and averaged the spiking probability across it and the two adjoining time bins. We computed the median probability of a spike occurring across all bins in the inter-pulse interval, to quantify the component of the response that was not pulse-locked. We computed the difference between these two values to create a phase-locking index. We sampled 20% of the PTAs for each motor and stimulation channel pair, shuffled the spike times, thus obtaining PTAs that were matched in spike count, and computed the same phase-locking index above for PTAs generated from the shuffled data. We repeated this shuffling procedure 5000 times to create a null distribution of pulse-locking indices. PTAs were considered to be significantly pulse-locked if the index was greater than that 99% of those obtained by chance (i.e., $p < 0.01$). We also estimated the latency and jitter of significantly pulse-locked responses. To this end, we randomly sampled 20% of the inter-pulse intervals and computed the PTA for this sample. We then identified the bin with the maximum spiking probability thus determining its latency. We repeated this procedure 5000 times to get a distribution of latencies, the mean and variance of which were the latency and jitter estimate for that stimulation/recording pair.

## Quantifying somatotopically mapped connectivity

We sought to determine whether motor channels that encode information about specific digit movements also respond to stimulation in somatosensory cortex that evokes a touch sensation on the same digit. To this end, participants performed an attempted digit movement task. On each trial, a digit was cued and the participant attempted to flex then extend the digit before the next digit was cued. Participant C1 was cued by the name of the digit being spoken, then attempted to move his own paralyzed digit in synchrony with a virtual hand (MuJoCo, DeepMind Technologies, London, UK) performing the same

instructed movement. He completed 125 trials of this task in one session. Participants P2 and P3 were cued by watching a set of 5 colored circles displayed on a monitor in front of them. The circles were arranged to mimic the distribution of digit tips resting open on a table or keyboard. When a circle was filled by a gray dot the participant would attempt to flex the corresponding digit until the gray dot disappeared. Following a chime, he then attempted to extend the same digit. Each participant completed 50 trials of this task.

*Motor maps.* To generate a map of digit selectivity across M1, we first computed the mean peri-event time histogram (in 20-ms bins) for each motor channel across a 2 s period centered on the start of movement for each digit flexion. From these, we then identified, for each motor channel, the response window during which the difference between the maximum response and the minimum response (each corresponding to flexion of different digits) was maximal. We used different time windows for different M1 channels because some units were most strongly active during preparation and others during movement. The modulation value for each digit was then calculated by subtracting the mean firing rate across all digits from the average firing rate for one digit, and then dividing by the mean firing rate across all digits. Plotting this modulation value across all channels for one digit provides a map of selective activation for that digit.

To generate *sensory projection maps*, we first computed the modulation values for each motor channel when stimulation was delivered through stimulation channels that evoked a sensation on the palmar side of a given digit. For example, we computed the modulation value for each motor channel when all the stimulation channels with projected fields on the thumb were stimulated. We then averaged these modulation values to obtain the thumb projection map. We repeated this procedure for all the digits (excluding the little finger for participant C1 and the ring and little finger for participant P3, because they never reported sensations there).

Reasoning that the motor maps and sensory projection maps reflect individually noisy estimates of the digit preference of individual motor channels, we convolved the maps with a 2D Gaussian whose standard deviation was equal to the spacing between two adjacent electrodes, to reinforce local patterns of digit preference. Note that the subsequent analyses were also performed without spatial filtering and yielded weaker but similar results.

To test for somatotopic linkage between S1 and M1, we first compared, for each motor channel, the activation evoked when ICMS was delivered through S1 channels whose projected field matched the digit that evoked the strongest response during attempted movement to the activation evoked when ICMS was delivered through S1 channels whose projected field did not match the movement field of the M1 channel. For this analysis, the ICMS-evoked activity was normalized within digit: For example, the activation on a given M1 channel evoked by ICMS through all S1 channels with projected fields on the thumb was normalized by the mean activation across all M1 channels evoked by ICMS through all S1 electrodes with projected fields on the thumb. This normalization was implemented to remove incidental digit-specific differences in the efficacy of stimulation array-wide. For example, thumb electrodes might more effectively drive stimulation across the array than index electrodes. We could then compare the activation on a given motor channel when ICMS was delivered through S1 channels with PFs that were predominantly on the digit that most strongly activated that motor channel to the activation evoked by non-matching S1 channels (using a Wilcoxon rank sum test). To visualize the array-wide somatotopic organization of the motor map, we calculated the Spearman correlation between the motor activation of each electrode and the corresponding digit (thumb = 1, index = 2, …, pinky = 5). Accordingly, channels that responded preferentially to the lateral digits (ring and little finger) yielded positive correlations; channels that responded preferentially to the medial digits (thumb and index) yielded negative ones. To visualize the array-wide somatotopic

organization of the sensory projection map, we calculated the Spearman correlation between ICMS-evoked activation by digit. Accordingly, M1 channels that were most activated by S1 channels with projected fields on the lateral digits yielded positive correlations; M1 channels that were preferentially activated by S1 channels with projected fields on the medial digits yielded negative correlations. The resulting maps revealed preference gradients across the arrays (Supplementary Fig. 11).

Next, we assessed whether the strength of the ICMS-evoked activity could be predicted from its digit preference profile. The activation of each M1 channel during attempted movement of each digit provided a digit preference profile for that channel's motor signals. The ICMS-evoked activation of individual M1 channels by stimulation across all channels with PFs on each digit provided a digit preference profile for that channel's S1 projections. We could then test whether these two profiles matched. For example, it is an M1 channel that responded most to D1 flexion, second most to D3 flexion, and third most to D2 flexion most activated by S1 channels with PFs on D1, then D3, then D2. First, we computed the mean activation on the digits, ordered by preference, to assess the mean effect across M1 channels. We also assessed the effect at the level of single motor channels by computing the Pearson correlation coefficient for the motor and sensory projection digit preference profiles. To assess significance, we computed a null distribution of correlations by shuffling both the electrode and digit assignments of the responses 10,000 times and computing the resulting correlations. Both the motor and sensory digit preference profiles were computed from the spatially smoothed motor and sensory projection maps. The null distributions were also computed after shuffling and then smoothing, to ensure that our findings were not artifacts of the smoothing.

P2 exhibited a different pattern of results than did C1 and P3. Given that P2's arrays had been implanted for much longer than C1's and P3's, we verified that the different pattern was not due to array malfunction. To this end, we measured the activity of P2's lateral motor array during a task requiring movement of the proximal arm muscles. We asked the participant to perform overt center-out planar reaches to eight targets (10 reaches per target) on a smooth surface, with his hand supported. On each trial, we cued the target location and reach timing on a screen in front of the participant. Each trial comprised a half-second presentation phase followed by an approximately one-and-a-half-second reach. To analyze the resulting neural data, we first binned threshold crossings into 20-ms bins and convolved these with a Gaussian kernel (std = 100 ms) to achieve a smooth estimate of the firing rate. We averaged these rates across repetitions for each target and normalized them so that each channel ranged from 0 (minimum firing rate) to 1 (maximum firing rate). For presentation purposes, we estimated the preferred direction of each channel using a cosine model and sorted the channels accordingly. We also confirmed that the modulation carried significant information by classifying the movement target. We first took the average firing rate of each channel on the lateral array during the reach phase of each trial. We then trained a linear discriminant analysis classifier on the top 10 principle components and tested it using leave-one-out cross-validation.

## Assessing the task dependence of ICMS-evoked activity in M1

We sought to determine whether the effects of ICMS to S1 on M1 activity depended on the task. To this end, we had participants C1 and P3 perform two tasks while we delivered ICMS to S1.

In the *squeeze task*, the participant squeezed a virtual object and reported the intensity of the ICMS-evoked touch sensation. On each trial the participant attempted to squeeze a virtual object with a medium amount of force, following the trajectory of a virtual hand observed through a VR headset. Upon contact with the object, stimulation was delivered on two electrodes at one of four amplitudes (20, 32, 44, or 56 µA). The hand continued to grasp the object for 1 s

before a release cue appeared. Once the hand released the object, the participant reported the perceived intensity of the stimulation using a scale of his choosing, with the following instructions. If he did not feel the stimulus, he ascribed to the sensation a rating of zero. If a stimulus on one trial felt twice as intense as that on another, he ascribed a rating that was twice as high (other such examples were provided). He was encouraged to use decimals or fractions. The main goal of the magnitude estimation component was to keep the participant engaged in the task.

In the *grasp and transport task*, an object appeared at one corner of an invisible cube centered on the starting point of the virtual hand. The participant then attempted to reach to the object, following the movement of the virtual hand. Once there, the participant attempted to grasp the object with medium force. During the grasp, ICMS was delivered at one of four amplitudes, as in the squeeze task. The participant then attempted to bring the object back to the center of the cube and release it there, again following the movements of the virtual limb.

Participant C1 completed 208 trials of each task in each of two sessions. Participant P3 completed 160 trials of each task in one session.

To confirm that the participant was attending to the grasp and transport task, we classified the intended target during the reach phase of the task. A naïve Bayes classifier was trained using 1 s of data from all active motor channels (>5 Hz mean firing rate across whole task) starting 400 ms before movement onset. This classifier was tested using leave-one-out cross-validation.

To assess whether the ICMS-evoked M1 activity varied across tasks, we analyzed the firing rates across all motor channels during three distinct phases across the two tasks: The 1-sec period after contact during the squeeze task; the 1-s period after contact during the grasp and transport task, and the first second of the transport phase in the grasp and transport task. In all three of these phases, the ICMS was identical but the movements were different (squeeze/grasp vs. transport) or their context was different (squeeze vs. grasp). We performed a multivariate ANOVA on the firing rates to determine which channels were significantly modulated by changes in task phase, stimulation level, and the interaction of the two. As an index of task dependence, we computed the ratio of the F-statistic for the interaction effect to that for the main effect of stimulation. This value was high to the extent that the interaction effect was strong compared to the main effect. This index was only computed for significantly modulated channels.

To assess whether task-dependent differences in susceptibility to stimulation reflected differences in task-dependent baseline firing rate, we investigated the relationship between ICMS-induced modulation and the baseline firing rate for each phase for each M1 channel. Baseline firing rates for the squeeze and grasp phases were calculated during the half second preceding object contact, during which time the hand was moving but no ICMS was delivered. The baseline firing rate for the transport phase was calculated using the first half-second during the reach phase, a similar movement without ICMS. The baseline firing rate in each phase for each neuron was then normalized by the mean baseline firing rate across phases. The index of modulation was the average firing rate during stimulation at the highest amplitude (56 μA) minus the average firing rate during stimulation at the lowest amplitude (20 μA) for each channel and phase, divided by the mean baseline firing rate across phases (computed as described above). We then plotted the modulation against the baseline firing rate for each channel and phase. To the extent that differences in modulation strength reflected a saturation effect, we expected a negative relationship between modulation strength and baseline firing rate.

To further determine how different the ICMS-evoked M1 activity was across tasks, we performed a linear discriminant analysis to identify stimulation amplitude based on the M1 activity during squeeze, grasp, and transport phases separately (during the half

second after contact initiation during the squeeze and grasp and during the first half second of transport), after subtracting the baseline activity for each phase (as described above) to remove task-dependent activity. These classifiers were tested on all three conditions. Within-condition accuracy was calculated using leave-one-out cross-validation, while cross condition accuracy was calculated using a decoder built from all available trials in each condition.

## Quantifying the impact of ICMS on motor decoding

We sought to determine whether the M1 activity evoked by ICMS would disrupt the ability of the participants to control a virtual arm. In 3 sessions, participant C1 attempted to make the movements of a virtual hand and arm displayed in his VR headset. On each trial, the virtual hand reached to an object, grasped it, transported it to a new location, and released it. After completing 60 trials, we trained a decoder for three-dimensional translation of the hand using these data. The decoder used throughout this project was an indirect Optimal Linear Estimator with ridge regression[9,15]. Next, we measured neuronal activity as the participant controlled translation, but with the computer preventing deviations from the path to the target (for an additional 60 trials). A new decoder was then trained from these data, and that decoder was used for the rest of the session. Throughout the session, the hand grasped automatically under computer control to ensure that stimulation was applied consistently across all trials while the participant controlled hand translation. The decoders were trained without stimulation but with blanking applied at 100 Hz during object contact to simulate the neuronal signal available during online decoding with ICMS-based feedback.

Once the decoder was trained, performance was tested under three conditions. In the "no stimulation" condition, the participant performed the same task that was used during training; no stimulation was provided but a 1.6-ms window of neuronal data was blanked at 100 Hz to match the data available during stimulation. In the two stimulation conditions, ICMS was delivered on two electrodes, one with projected fields on the thumb and one with projected fields on the index finger. In the linear condition, the ICMS frequency was 100 Hz and the amplitude was 52 μA. In the "biomimetic stimulation" condition, 100-Hz ICMS comprised onset and offset transients at 72 μA for 200 ms and sustained stimulation at 32 μA during maintained contact. The order of the test blocks was randomized in each session, with each condition used for two sets of ten trials before the next condition was tested. Conditions were repeated three times to obtain a total of 60 trials for each.

If the participant was unable to place the hand at the target location within 10 s during either the reach or the transport phase, the trial was terminated and marked as a failure. To determine the causes of failure, we computed the path length during the transport phase (when stimulation was provided and the participant had control of the arm) for every trial, even if the trial failed during that phase. The median path lengths were compared across stimulation conditions using the Wilcoxon rank-sum test to determine significance.

## Reporting summary

Further information on research design is available in the Nature Portfolio Reporting Summary linked to this article.

# Data availability

The deidentified data generated in this study have been deposited in the Data Archive BRAIN Initiative (DABI) under project code 6281M47AHII3. The data are available under restricted access for participant privacy, access can be obtained upon request to the study PIs by an investigator who is prepared to securely handle data resulting from human research. Source data are provided with this paper.

## Code availability

Custom code used for analysis is available through Github (https://doi.org/10.5281/zenodo.8408314). Code used for data collection can be made available upon request to the study PIs.

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

## Acknowledgements

This work was supported by NINDS grants UH3 NS107714 and R35 NS122333.

## Author contributions

Conceptualization: N.D.S., J.E.D., and S.J.B.; Development: N.D.S., J.E.D., and C.M.G.; Investigation: N.D.S., J.E.D., C.M.G., E.V.O., A.R.S., C.V., Q.H., A.F.T., D.D.M., L.E.M., R.A.G., J.L.C., and N.G.H.; Analysis: N.D.S., J.E.D., C.M.G., E.V.O., A.R.S., M.T.K., R.A.G., and S.J.B.; Clinical Oversight: R.C.L., D.S., J.G.M., and P.C.W.; Supervision: M.L.B., R.A.G., J.L.C., N.G.H., and S.J.B.; Writing: N.D.S., J.E.D., C.M.G., and S.J.B.; Review and Editing: All Authors.

## Competing interests

N.H. and R.G. serve as consultants for Blackrock Microsystems, Inc. R.G. is also on the scientific advisory boards of Braingrade GmbH and Neurowired LLC. M.B., J.C., and R.G. received research funding from Blackrock Microsystems, Inc. though that funding did not support the work presented here. The remaining authors declare no competing interests.
