## [Peer Review File · Nature Communications]

Reviewers' Comments:

Reviewer #1:

Remarks to the Author:

This study investigates the functional relationship between upper limb representations in the primary somatosensory (SI) and motor (MI) cortices of 3 human participants. To this end, the authors analyzed the unitary and multi-unit activity of the motor cortex in response to trains of intracortical microstimulation (ICMS) applied to the somatosensory cortex through high density multi-electrode arrays. Three main results emerge from the analysis of the ICMS evoked responses in MI. First, the spatial distribution of these evoked responses is modulated by the location of the stimulating electrode in SI, with a close match between the individual digit representations in these two cortical areas. Second, the amplitude of evoked responses is dependent on the behavioral task in which the animal is engaged. And third, the decoding of motor intent from MI activity is degraded by the ICMS protocol, but this negative effect of ICMS is suppressed when the stimuli used have biomimetic characteristics. Overall, the issues addressed in this study are relevant and the experimental approach is adequate. The first result concerning the response properties of the neurons and their spatial distribution does not seem too surprising based on the older literature in the field (see comments below). The modulation of the ICMS effects evoked by the behavioral context is very interesting but its analysis needs to be clarified. The analysis of the interference between ICMS and decoding is undoubtedly the most interesting part because it validates the possibility of using SI ICMS in conjunction with BMIs for movement control. Therefore from my point of view, a few points in the study need further investigation or clarification before this work can be considered for publication in nature communication.

The end of the introduction might suggest that no previous studies have explored the response of unitary MI neurons to SI stimulation. It would be essential to mention studies that have investigated the relationship between SI and MI in great details in cats and monkeys using closely related approaches. Notably, Ghosh and Porter (1988) made intracellular recordings of MI neurons and analyzed their responses to SI stimulation. They precisely describe the distributions of response latencies and intensity effects. Anatomically, the projections between SI and MI are well identified in non-human primates (Ghosh et al., 1987; Tokuno and Tanji; 1993) and it has been suggested that they preferentially connect cortical zones with similar body representation in SI and MI, in agreement with the first result of the current work. I advise the authors to do a thorough reading of this literature in order to clarify the originality of their approach and to enrich the discussion of their results.

To quantitatively assess the somatotopic organization of SI to MI projections, a 2D Gaussian smoothing is applied on the sensory and motor maps before computing the covariance. Did the author also apply the same smoothing on the 2D scrambled maps before computing the covariance between maps? The spatial smoothing is expected to substantially increase the covariance values in the null distribution.

Concerning the modulation of stimulation evoked response across tasks, the authors examined first the task dependence of M1 activity. Did the authors controlled whether the levels of background activity were comparable between the three tasks? It is indeed likely that in these virtual reality tasks, modulation of activity are not precisely timed with the video and global modulation may occur before the virtual contact with the object. If this is the case, would it be possible that the largest responses to SI ICMS are evoked in the task in which the units are showing the highest level of background activity?

Minor

How long after array implantation did the ICMS protocol take place? Did the protocol occur roughly at the same time in the 3 participants. In particular with these chronic implants, it is likely that the efficacy of ICMS decreases with time.

The first part of the analysis presents the distribution of ICMS-driven modulation of activity for each participant, normalized by baseline fluctuation. I assume that figure 2B includes all the significant and non-significant responses and combines excitatory and inhibitory effects. As indicated in the methods, "Positive modulation values indicate an excitatory effect while negative modulation values indicate an inhibitory effect." In line with this observation, I actually find that figure S4 is more informative than figure 2B to illustrate the large range of responses in the 3 participants.

Is the sign of the evoked responses always consistent on a given M1 channel, independent of the

SI stimulating channel?

The cumulative latency distributions in figure 3B include both direct and indirect responses. Yet, it is not clear from figures 3C and S8 if the latency of response makes really sense for indirect responses. Not surprisingly for subject P2 in which 99,4% of the responses are indirect, the latency distribution reach a plateau at very short latency. I think it would make more sense to show these distributions only for pulse-locked responses in subjects C1 and P3.

For putative indirect responses like the one illustrated in fig 3C, is there any buildup of activity across pulses, which would support the transynaptic nature of the responses. Alternatively, could it be that these responses correspond to the tail of very short latency response with a peak hidden by the blanking?

What does the color scale represent in figure 4A, changes in FR during the stimulation period?

The spatial layout of figure 5 is unclear. It is difficult to understand what exactly is the top part of 5B or the bottom part of 5A. The authors should maybe use additional letters in the legend.

The blanking of the stimulation artefact is 2 ms long and probably obscures most of the very short latency effects observed by Gosh and Porter. The authors should discuss this point.

Reviewer #2:

Remarks to the Author:

Title

MICROSTIMULATION OF HUMAN SOMATOSENSORY CORTEX EVOKES TASK-DEPENDENT, SPATIALLY PATTERNED RESPONSES IN MOTOR CORTEX
Summary

The study examines the effects of applying intracortical microstimulation in primary somatosensory cortex (S1) on the modulation of neurons in primary motor cortex (M1) in three human participants implanted with intracortical microelectrode arrays. Given that there are S1-M1 projections, the study aims to quantify at the single neuron resolution how M1 modulation varies wrt stimulation site and task. The primary claimed takeaways from the study are that there is significant overlap in representations of individual fingers in M1 during motor tasks and as a result of S1 stimulation (meaning stimulation of the portion of the S1 array that results in tactile percepts of a specific hand part results in modulation of the M1 array of that same representative location). The modulation responses were found to be task specific and vary systematically across the stimulated electrode. S1 stimulation was found to disrupt motor decoding from M1, as would be expected, but the motor decoding could be restored using a biomimetic stim pattern rather than a continuous stim pattern.

General Comments (Major)

The manuscript presents a number of observations about what happens in M1 during S1 stimulation, but with little meaningful interpretation to connect the various results, and with some results and conclusions not really supported by the data. The main takeaway, that there is meaningful somatotopic overlap between the M1 array activation during a motor task and that which occurs during S1 stimulation, while perhaps evident in figure 4 for participant C1, seems to not be supported by Supplementary figure 11 which shows the data for all participants and all fingers. The premise seems to hold for thumb modulation for C1, and to a lesser degree for ring, but the activations for the remaining fingers seem to be so diffuse and widespread that it is unclear if the result is due to overlapping somatotopy, particularly when the hotspots are not aligned. This is even more starkly evident when looking at panels B and C showing the results of participants P2 and P3. To make a general conclusion based upon one participant's partial responses and it not be supported by the other 2 participants does not seem appropriate. Might the authors be able to offer some reasonable explanation for why they see this phenomenon in only 1 of 3 participants, and yet choose to come to this conclusion?

The various observations (task dependency, phase locking, somatotopy, excitatory vs inhibitory responses to stimulation), while all are interesting, are not connected together into a cogent story, which seemingly should be about the fundamental connectivity of S1 to M1 at the single unit level.

Thus, it is difficult to know what to actually take away from this paper that influences our understanding of cortical organization. The interesting result of using biomimetic stimulation to mitigate the negative effects of S1 stimulation during M1 motor decoding is interesting, but it seems to awkwardly fit with the rest of the study since it does not seem to address what again is presumably the main thesis of M1-S1 connectivity. Biomimetic vs continuous stimulation seems to be more about a perceptual response and its usefulness in BCI control. How does the type of stimulation help in understanding M1-S1 connectivity?

Specific Comments

Lines 153-156: Can you quantify the discriminability of these different spatial patterns? Your only analysis is that they are different than randomly shuffled data sets. But if your argument is that they are somatotopically dependent on which electrodes were stimulated, please provide some quantification of the electrode dependent differences in stimulation patterns.

Lines 264-280: What is the implication of this?

Lines 345-346: it is unclear what it means that the signal transmitted are the same across tasks but the indirect effects are different. What is the underlying cause of the different effects of the same signals?

Lines 477-479: Can you comment if this was hardware or software based blanking?

Numerous typos, as well as "(Error! Reference source not found)" all throughout the manuscript.

Reviewer #3:

Remarks to the Author:

The prepared manuscript by Shelchkova et al. provides data from clinical trials of microstimulation. Three human participants were subjected to stimulation of the primary somatosensory cortex (S1) and the subsequent electrophysiological responses were recorded on microelectrode arrays implanted in the primary motor cortex (M1). They report both direct and indirect activation of primary motor cortex. Further, the data, especially the direct activation data, demonstrated a principled arrangement according to somatotopy - electrodes in a given primary S1 somatotopic location evoked neuronal activity on electrodes implanted in the M1 location responsible for movement of that somatotopic location. Interestingly, the indirect activation data was variable depending on context. The authors use this knowledge to design and test S1 stimulation encoding algorithms. The results are interesting and provide a significant contribution to the field of neuroprostheses. Furthermore, they add to the paltry cellular neurophysiology knowledge of human cortex. However, there are still opportunities to improve the presentation of the information and data.

Major comments:

- The term biomimetic is used without definition. In strictest terms, biomimetic indicates the identical mimicry of the biology and physiology. This reviewer agrees that phasic stimulation trains that emphasize contact transients and reduce sustained stimulation are more naturalistic. However, ICMS trains delivered from extracellular electrodes evoking activity from dozens of various neural types simultaneously (not to mention glia) is not identical mimicry of the biology and physiology. A comment of clarification is needed to define biomimetic as the authors intend its understanding.
- Figure caption 1 needs clarification. It is not clear what, "Top two from participant P3, bottom four from participant C1." means. There are not 6 panels.
- Figure caption 4 needs clarification. To describe the bottom sub-panel of Figure 4A it says, "Difference in activation during attempted flexion of the thumb (left) and ring finger (right) vs. the mean activation during attempted flexion of each of the 5 digits." However, the figure shows a heat-map. How does a heat-map represent attempted flexion of each of the 5 digits?
 - o Furthermore, the heatmap description in the figure 4B caption is very confusing. For example, the caption says that the top heatmap on the left side of 4B represents the "Average M1 activity

evoked by stimulation through S1 channels with projected fields on the thumb." However, it appears that the heatmap is null.

Minor comments:

Figure 3 caption: The hyphen should be removed from 100-Hz.

We would like to thank the reviewers for their thoughtful comments. We have made extensive changes throughout the manuscript, as detailed below. In particular, we replaced the analysis that demonstrated a somatotopic link from S1 to M1 with three much more straightforward analyses that yield the same conclusion. We also endeavored to link the different results together to improve the flow. We feel the paper is much improved from these changes and hope you will too.

Reviewer #1

This study investigates the functional relationship between upper limb representations in the primary somatosensory (SI) and motor (MI) cortices of 3 human participants. To this end, the authors analyzed the unitary and multi-unit activity of the motor cortex in response to trains of intracortical microstimulation (ICMS) applied to the somatosensory cortex through high density multi-electrode arrays. Three main results emerge from the analysis of the ICMS evoked responses in MI. First, the spatial distribution of these evoked responses is modulated by the location of the stimulating electrode in SI, with a close match between the individual digit representations in these two cortical areas. Second, the amplitude of evoked responses is dependent on the behavioral task in which the animal is engaged. And third, the decoding of motor intent from MI activity is degraded by the ICMS protocol, but this negative effect of ICMS is suppressed when the stimuli used have biomimetic characteristics. Overall, the issues addressed in this study are relevant and the experimental approach is adequate. The first result concerning the response properties of the neurons and their spatial distribution does not seem too surprising based on the older literature in the field (see comments below). The modulation of the ICMS effects evoked by the behavioral context is very interesting but its analysis needs to be clarified. The analysis of the interference between ICMS and decoding is undoubtedly the most interesting part because it validates the possibility of using SI ICMS in conjunction with BMIs for movement control. Therefore, from my point of view, a few points in the study need further investigation or clarification before this work can be considered for publication in nature communication.

The end of the introduction might suggest that no previous studies have explored the response of unitary MI neurons to SI stimulation. It would be essential to mention studies that have investigated the relationship between SI and MI in great details in cats and monkeys using closely related approaches. Notably, Ghosh and Porter (1988) made intracellular recordings of MI neurons and analyzed their responses to SI stimulation. They precisely describe the distributions of response latencies and intensity effects. Anatomically, the projections between SI and MI are well identified in non-human primates (Ghosh et al., 1987; Tokuno and Tanji; 1993) and it has been suggested that they preferentially connect cortical zones with similar body representation in SI and MI, in agreement with the first result of the current work. I advise the authors to do a thorough reading of this literature in order to clarify the originality of their approach and to enrich the discussion of their results.

Thank you for bringing these studies to our attention. The Ghosh study reveals short-latency responses in monkey M1 triggered by surface stimulation of S1. The Tanji study provides an anatomical basis, again in monkeys, for the somatotopic linkage that we observe. Our study builds on these by (1) demonstrating the effect in humans; (2) characterizing the incidence of the activity evoked by S1 microstimulation across motor cortex; (3) showing the functional correlates of the somatotopic linkage; and (4) exploring the implications for BCIs. We now explicitly discuss our results in the context of this previous work.

To quantitatively assess the somatotopic organization of SI to M1 projections, a 2D Gaussian smoothing is applied on the sensory and motor maps before computing the covariance. Did the author also apply the same smoothing on the 2D scrambled maps before computing the covariance between maps? The spatial smoothing is expected to substantially increase the covariance values in the null distribution.

Excellent question. We did, indeed, apply the same procedure to the scrambled maps. In the revised manuscript, in response to comments by the other reviewers, we have replaced that analysis with different (more straightforward) ones.

Concerning the modulation of stimulation evoked response across tasks, the authors examined first the task dependence of M1 activity. Did the authors controlled whether the levels of background activity were comparable between the three tasks? It is indeed likely that in these virtual reality tasks, modulation of activity are not precisely timed with the video and global modulation may occur before the virtual contact with the object. If this is the case, would it be possible that the largest responses to SI ICMS are evoked in the task in which the units are showing the highest level of background activity?

While the levels of background activity differ across tasks, there is no systematic relationship between the effect of S1 stimulation on M1 activity and the baseline activity. Indeed, the effect of task on S1-evoked M1 activity differs across S1-M1 electrode pairs. This is now clarified in the following passage:

“The task dependence of ICMS-evoked M1 activity varied idiosyncratically across M1 channels. For example, one M1 channel was susceptible during the transport phase but not the grasp phase while another channel showed the reverse pattern of responses (Figure 6A). This heterogeneity suggests that the task-dependence of the modulation was not driven by array-wide differences in the baseline activity across conditions, for example, reflecting saturation due to higher baseline activation in some conditions. Even at the single channel level, the modulation strength was not systematically related to the (task-dependent) baseline firing rate ($p > 0.05$ for both participants, Friedman’s Test, Supplementary Figure 16). This heterogeneous pattern of task-dependent susceptibility to ICMS implies that ICMS-evoked activity in M1 cannot be straightforwardly distinguished from intrinsic motor-related activity.”

Minor

How long after array implantation did the ICMS protocol take place? Did the protocol occur roughly at the same time in the 3 participants. In particular with these chronic implants, it is likely that the efficacy of ICMS decreases with time.

Sensitivity to ICMS has been shown to be highly stable over time in both monkeys (Callier et al. JNE, 2015;) and humans (Hughes et al., JNE, 2021). C1 and P3 received their implants at similar times (~1.5-2 years prior to these experiments) whereas P2 received his much earlier (~6.5 years prior to these experiments) but, given P2’s high sensitivity to ICMS, we do not believe this difference drives the differences between C1/P3 and P2. We added information about time since implant to the methods.

The first part of the analysis presents the distribution of ICMS-driven modulation of activity for each participant, normalized by baseline fluctuation. I assume that figure 2B includes all the significant and non-significant responses and combines excitatory and inhibitory effects. As indicated in the methods, “Positive modulation values indicate an excitatory effect while negative

modulation values indicate an inhibitory effect.” In line with this observation, I actually find that figure S4 is more informative than figure 2B to illustrate the large range of responses in the 3 participants.

The main message of panel 2B is to convey the preponderance of the effect. A figure that preserves the sign of the effect obscures its preponderance. Accordingly, we would prefer to keep the figures as they are. All of the relevant information is available. We do reference Figure s4 in this paragraph and describe the distribution of excitatory and inhibitory responses for each participant in the main text. “The participants also differed in the sign of the ICMS-induced modulation, with primarily excitatory responses in C1 (94.2%) and a more even mix in P2 and P3 (39.3% and 46.0% excitatory, respectively).”

Is the sign of the evoked responses always consistent on a given M1 channel, independent of the SI stimulating channel?

We added the following passage to address this excellent question:

“Most modulated M1 channels exhibited both increases and decreases in ICMS-evoked activity, depending on the stimulation channel (48%, 90%, and 98% for C1, P2 and P3 respectively).”

The cumulative latency distributions in figure 3B include both direct and indirect responses. Yet, it is not clear from figures 3C and S8 if the latency of response makes really sense for indirect responses. Not surprisingly for subject P2 in which 99,4% of the responses are indirect, the latency distribution reach a plateau at very short latency. I think it would make more sense to show these distributions only for pulse-locked responses in subjects C1 and P3. For putative indirect responses like the one illustrated in fig 3C, is there any buildup of activity across pulses, which would support the transynaptic nature of the responses. Alternatively, could it be that these responses correspond to the tail of very short latency response with a peak hidden by the blanking?

The latencies are only reported for pulse locked responses. The effects last throughout the entire stimulation period but sometimes decrease, as shown in the rasters in Figure 1 and Supplementary Figure 2.

What does the color scale represent in figure 4A, changes in FR during the stimulation period?

The caption was updated to read “Z-scored difference in firing rate during attempted flexion of the thumb (left) and ring finger (right) vs. the mean activation during attempted flexion of each of the 5 digits.”

The spatial layout of figure 5 is unclear. It is difficult to understand what exactly is the top part of 5B or the bottom part of 5A. The authors should maybe use additional letters in the legend.

Thank you for pointing out the vagueness of the labeling. Additional labels were added for clarity (in what is now Figure 4).

The blanking of the stimulation artefact is 2 ms long and probably obscures most of the very short latency effects observed by Gosh and Porter. The authors should discuss this point.

We added the following passage in the Discussion:

“In macaques, surface stimulation of S1 was shown to evoke responses in M1 with latencies ranging from ~1 to 7 ms⁸, consistent with our results. Because we discard the first 2 ms of the

response after pulse onset to avoid contamination from the stimulation artifact, we likely missed some responses that occurred at shorter latencies.”

Reviewer #2

Summary

The study examines the effects of applying intracortical microstimulation in primary somatosensory cortex (S1) on the modulation of neurons in primary motor cortex (M1) in three human participants implanted with intracortical microelectrode arrays. Given that there are S1-M1 projections, the study aims to quantify at the single neuron resolution how M1 modulation varies wrt stimulation site and task. The primary claimed takeaways from the study are that there is significant overlap in representations of individual fingers in M1 during motor tasks and as a result of S1 stimulation (meaning stimulation of the portion of the S1 array that results in tactile percepts of a specific hand part results in modulation of the M1 array of that same representative location). The modulation responses were found to be task specific and vary systematically across the stimulated electrode. S1 stimulation was found to disrupt motor decoding from M1, as would be expected, but the motor decoding could be restored using a biomimetic stim pattern rather than a continuous stim pattern.

General Comments (Major)

The manuscript presents a number of observations about what happens in M1 during S1 stimulation, but with little meaningful interpretation to connect the various results, and with some results and conclusions not really supported by the data. The main takeaway, that there is meaningful somatotopic overlap between the M1 array activation during a motor task and that which occurs during S1 stimulation, while perhaps evident in figure 4 for participant C1, seems to not be supported by Supplementary figure 11 which shows the data for all participants and all fingers. The premise seems to hold for thumb modulation for C1, and to a lesser degree for ring, but the activations for the remaining fingers seem to be so diffuse and widespread that it is unclear if the result is due to overlapping somatotopy, particularly when the hotspots are not aligned. This is even more starkly evident when looking at panels B and C showing the results of participants P2 and P3. To make a general conclusion based upon one participant's partial responses and it not be supported by the other 2 participants does not seem appropriate. Might the authors be able to offer some reasonable explanation for why they see this phenomenon in only 1 of 3 participants, and yet choose to come to this conclusion?

The phenomenon is most visually striking in participant C1 but is also observed in participant P3. The analysis was confusing and unconvincing so we performed new analyses, which yielded the same conclusion. First, we showed that stimulating channels with projected fields on the digit that most activates a channel in the motor task leads to higher activation than stimulation of channels with non-matching projected fields (New Fig 5A). Second, we showed that, on average, this activation drops as expected depending on the projected field of the stimulating channel. If the projected field matches the preferred digit of the motor channel, the activation is highest. If the projected field matches the second preferred digit of the motor channel, its activation is second highest, and so on... (Fig 5B). Third, the pattern of digit preference of a motor channel is predictive of the degree to which individual stimulating channels can activate it, based on the latter's pattern of digit preference (Supp Fig 12). These analyses are much more straightforward and lead to the conclusion of that S1 and M1 are somatotopically linked.

As for P2, we state the following in the manuscript

“The somatotopic link between S1 and M1 was much weaker and, in fact, non-significant in participant P2. Note, however, that ICMS-driven M1 activation in this participant was sparse (Figure 2A), weak (Figure 2B), and unpatterned (Supplementary Figure 9C,D). We attribute the lack of spatial patterning to the fact that this participant's most lateral M1 array, more medial than its counterparts in the other two participants, was located in the proximal limb representation as evidenced by robust arm- but weak digit-related activity (Supplementary Figure 11B). The M1 arrays in participant P2 are also much older than are those in participants C1 and P3, which may have contributed to the observed differences, though robust movement-related signals could still be harnessed from this array (Supplementary Figure 13).”

The various observations (task dependency, phase locking, somatotopy, excitatory vs inhibitory responses to stimulation), while all are interesting, are not connected together into a cogent story, which seemingly should be about the fundamental connectivity of S1 to M1 at the single unit level. Thus, it is difficult to know what to actually take away from this paper that influences our understanding or cortical organization. The interesting result of using biomimetic stimulation to mitigate the negative effects of S1 stimulation during M1 motor decoding is interesting, but it seems to awkwardly fit with the rest of the study since it does not seem to address what again is presumably the main thesis of M1-S1 connectivity. Biomimetic vs continuous stimulation seems to be more about a perceptual response and it's usefulness in BCI control. How does the type of stimulation help in understanding M1-S1 connectivity?

In the present study, we sought to document the S1-M1 connectivity phenomenon and assess its implications for BCI. We feel that the extensive characterization of the phenomenon helps frame its implication for BCIs and yields a more impactful paper. We have edited the text to make for smoother transitions between different sections.

Specific Comments

Lines 153-156: Can you quantify the discriminability of these different spatial patterns? Your only analysis is that they are different than randomly shuffled data sets. But if your argument is that they are somatotopically dependent on which electrodes were stimulated, please provide some quantification of the electrode dependent differences in stimulation patterns.

As mentioned above, we have taken these points seriously and completely changed the analysis that demonstrate the somatotopic linkage (see Figure 5).

Lines 264-280: What is the implication of this?

We have added the following passage to address this fair comment:

“This heterogeneous pattern of M1's susceptibility to ICMS implies that it cannot be accounted or compensated for using a linear model.”

Lines 345-346: it is unclear what it means that the signal transmitted are the same across tasks but the indirect effects are different. What is the underlying cause of the different effects of the same signals?

We clarify the implication by editing the sentence to read:

“In other words, the signals that are directly transmitted from S1 to M1 are consistent across tasks, but their indirect effects are not. The activity in M1 that is indirectly evoked by ICMS in S1 is highly task-dependent and the task dependence varies widely across M1 channels.”

Lines 477-479: Can you comment if this was hardware or software based blanking?

Thank you for pointing out this imprecision. This section was edited to read: “For these analyses, we discarded the first 2 ms after each pulse to eliminate any potential electrical artifacts that extended beyond the initial 1.6 ms blanking window.”

Numerous typos, as well as “(Error! Reference source not found)” all throughout the manuscript.

Fixed.

Reviewer #3

The prepared manuscript by Shelchkova et al. provides data from clinical trials of microstimulation. Three human participants were subjected to stimulation of the primary somatosensory cortex (S1) and the subsequent electrophysiological responses were recorded on microelectrode arrays implanted in the primary motor cortex (M1). They report both direct and indirect activation of primary motor cortex. Further, the data, especially the direct activation data, demonstrated a principled arrangement according to somatotopy - electrodes in a given primary S1 somatotopic location evoked neuronal activity on electrodes implanted in the M1 location responsible for movement of that somatotopic location. Interestingly, the indirect activation data was variable depending on context. The authors use this knowledge to design and test S1 stimulation encoding algorithms. The results are interesting and provide a significant contribution to the field of neuroprostheses. Furthermore, they add to the paltry cellular neurophysiology knowledge of human cortex. However, there are still opportunities to improve the presentation of the information and data.

Major comments:

The term biomimetic is used without definition. In strictest terms, biomimetic indicates the identical mimicry of the biology and physiology. This reviewer agrees that phasic stimulation trains that emphasize contact transients and reduce sustained stimulation are more naturalistic. However, ICMS trains delivered from extracellular electrodes evoking activity from dozens of various neural types simultaneously (not to mention glia) is not identical mimicry of the biology and physiology. A comment of clarification is needed to define biomimetic as the authors intend its understanding.

We define biomimetic in the following passages:

“Importantly, because ICMS-evoked activity in M1 is task dependent, its influence on a decoder cannot be easily predicted and eliminated. However, we reasoned that reducing the intensity of ICMS would reduce its deleterious effects. With peripheral nerve interfaces, biomimetic somatosensory feedback – characterized by high-amplitude phasic stimulation at the onset and offset of contact, and far weaker stimulation during maintained contact^{11,12} – has been shown to elicit more natural and intuitive sensations^{13,14}.”

“However, this disruption can be minimized with a more biomimetic form of somatosensory feedback, which emphasizes the transient phases of object contact (its onset and offset) and minimizes sustained ICMS, mirroring the patterns of neuronal activity during object interactions.”

We expanded the following passage to explain how the emphasis of contact transients is biomimetic:

“Fortunately, we were able to eliminate the impact of ICMS on decoding by implementing phasic biomimetic feedback, designed to mimic natural cutaneous responses in cortex. Indeed, throughout the somatosensory neuraxis, neural populations respond more strongly at the onset and offset of object contact and much more weakly to maintained contact 31,32. Sensory feedback with this property entails weaker ICMS during maintained grasp, thereby resulting in weaker and thus less disruptive effects of ICMS in M1 (Supplementary Figure 12).”

Figure caption 1 needs clarification. It is not clear what, "Top two from participant P3, bottom four from participant C1." means. There are not 6 panels.

We clarified the caption by changing the description to: “The green rasters are from participant P3, the rest from participant C1.”

Figure caption 4 needs clarification. To describe the bottom sub-panel of Figure 4A it says, "Difference in activation during attempted flexion of the thumb (left) and ring finger (right) vs. the mean activation during attempted flexion of each of the 5 digits." However, the figure shows a heat-map. How does a heat-map represent attempted flexion of each of the 5 digits? o Furthermore, the heatmap description in the figure 4B caption is very confusing. For example, the caption says that the top heatmap on the left side of 4B represents the "Average M1 activity evoked by stimulation through S1 channels with projected fields on the thumb." However, it appears that the heatmap is null.

This caption was clarified as follows:

“B| Normalized difference in firing rate during attempted flexion of the thumb (left) and ring finger (right) vs. the mean activation during attempted flexion of each of the 5 digits.”

Minor comments:

Figure 3 caption: The hyphen should be removed from 100-Hz.

Done.

Reviewers' Comments:

Reviewer #1:

Remarks to the Author:

I thank the authors for responding to my comments and improving the manuscript. As far as I am concerned, two points still need to be clarified before this study can be published.

- In the significance statement and with respect to earlier studies, I still think that the sentence L61 "the nature of signaling between them is not known" is an overstatement and needs to be more moderate. The issue of SI-MI communication has been addressed in earlier studies in which partial knowledge has been acquired. I don't think the present study takes the issue from unknown to known, but it does provide some new insights.

- Regarding the dependency of ICMS evoked response to background activity across tasks (L245), the methodology is still not fully clear to me. My question was whether the discharge frequency of M1 neurons just before stimulation onset (i.e. baseline activity) correlates with the intensity of the response to SI stimulation while the subjects are performing the tasks. More globally, could different levels of background firing rates in the three tasks partially explain the effects shown in figure 6?

In the first part of the manuscript (Gauging the strength of ICMS-driven activity in motor cortex), the authors analyzed ICMS effects by calculating the change in firing rate between a 1 second interval preceding the stimulation train and the firing rate during the stimulation train itself (L518). However if I understand correctly, it seems that to assess the task effects, the authors only compare the firing rate during the stimulation period (L656). Why in the latter analysis is the intensity of the response not related to the firing rate that precedes the stimulation?

I don't really understand how the controlled analysis presented in supplementary Figure 16 was actually performed to address this point. L662, the authors indicate that they "investigated the relationship between ICMS-induced modulation and mean firing rate during each phase for each M1 channel". Does it mean that the mean firing rate during each phase for each M1 channel includes the stimulation period? Would it not be possible to compare the discharge frequency just before the stimulation onset in the 3 tasks? I think this issue is critical for the mechanistic interpretation of the task related effects (which are currently not addressed in the general discussion)? Are these effects reflecting a higher neuronal excitability of M1 neurons due to their task related activity or are they reflecting changes at the stimulation site?

Reviewer #2:

Remarks to the Author:

The resubmission generally does a decent job of addressing the previous reviewers' comments. However, it is of concern that the manuscript resubmission removed significant portions of Supplementary Figure 11 that were in the first submission (specifically the sensory projection maps, which in my opinion, showed data that was not just inconclusive, but perhaps even cast doubt on the overall claim of strong alignment between the M1 and S1 fields.). Rather than simply removing that perhaps inconvenient and hard to explain data (namely, that for 2 of 3 subjects, the M1 and S1 fields really did not overlap, nor even have visually similar centroids of spatial activation), that particular strong claim of the manuscript should be reinterpreted. The caveats given for P2 (array age, lack of S1 activation) are understandable, but given the previously presented S1 and M1 maps of the previous Supplementary Figure 11 (which seems to be a much more direct analysis of the shared somatotopy data than the new Figure 5), it is not clear that the conclusion of overlapping fields is as strong as impressed by the manuscript.

Reviewer #3:

None

We would like to thank the reviewers for their insight and diligence in reviewing this manuscript, which has a lot of parts to it. This was a gargantuan task! We have addressed the remaining concerns by adding two new analyses. The first, shown in Supplementary Figure 16, demonstrates that the task-dependence of the ICMS-modulation of M1 activity is not an artefact of response saturation, as requested by Reviewer 1. The second, included to address a concern expressed by Reviewer 2 and shown in Supplementary Figure 11, provides a more complete picture of the digit dominance gradients in the motor and sensory projection maps and thus a better visualization of the somatotopic linkage than did the old figure, which only showed the dominant digit. These were excellent points, and we feel the paper is much improved by having addressed them. We hope you will too.

Reviewer #1 (Remarks to the Author):

I thank the authors for responding to my comments and improving the manuscript. As far as I am concerned, two points still need to be clarified before this study can be published.

- In the significance statement and with respect to earlier studies, I still think that the sentence L61 "the nature of signaling between them is not known" is an overstatement and needs to be more moderate. The issue of SI-MI communication has been addressed in earlier studies in which partial knowledge has been acquired. I don't think the present study takes the issue from unknown to known, but it does provide some new insights.

The sentence has been updated to read "Motor (M1) and somatosensory (S1) cortices play critical roles in motor control but the nature of the signaling between them remains to be conclusively elucidated."

- Regarding the dependency of ICMS evoked response to background activity across tasks (L245), the methodology is still not fully clear to me. My question was whether the discharge frequency of M1 neurons just before stimulation onset (i.e. baseline activity) correlates with the intensity of the response to SI stimulation while the subjects are performing the tasks. More globally, could different levels of background firing rates in the three tasks partially explain the effects shown in figure 6? In the first part of the manuscript (Gauging the strength of ICMS-driven activity in motor cortex), the authors analyzed ICMS effects by calculating the change in firing rate between a 1 second interval preceding the stimulation train and the firing rate during the stimulation train itself (L518). However if I understand correctly, it seems that to assess the task effects, the authors only compare the firing rate during the stimulation period (L656). Why in the latter analysis is the intensity of the response not related to the firing rate that precedes the stimulation?

I don't really understand how the controlled analysis presented in supplementary Figure 16 was actually performed to address this point. L662, the authors indicate that they "investigated the relationship between ICMS-induced modulation and mean firing rate during each phase for each M1 channel". Does it mean that the mean firing rate during each phase for each M1 channel includes the stimulation period? Would it not be possible to compare the discharge frequency just before the stimulation onset in the 3 tasks? I think this issue is critical for the mechanistic interpretation of the task related effects (which are currently not addressed in the general discussion)? Are these effects reflecting a higher neuronal excitability of M1 neurons due to their task related activity or are they reflecting changes at the stimulation site?

We had reasoned that the modulation to ICMS would decrease as the response to both movement and ICMS increased if the task-dependence reflected response saturation. In the revised document, we have tested the effect more directly by assessing the relationship between the modulation effect and baseline firing rate, as requested. We find that the modulation is unrelated to the baseline response (reflecting only the motor component, without ICMS). This new analysis is shown in Supplementary Figure 16 and described in the Methods as follows:

"To assess whether task-dependent differences in susceptibility to stimulation reflected differences in task-dependent baseline firing rate, we investigated the relationship between ICMS-induced modulation and the baseline firing rate for each phase for each M1 channel. Baseline firing rates for the squeeze and grasp phases were calculated during the half second preceding object contact, during which time the hand was moving but no ICMS was delivered. The baseline firing rate for the transport phase was

calculated using the first half-second during the reach phase, a similar movement without ICMS. The baseline firing rate in each phase for each neuron was then normalized by the mean baseline firing rate across phases. The index of modulation was the average firing rate during stimulation at the highest amplitude (56 μ A) minus the average firing rate during stimulation at the lowest amplitude (20 μ A) for each channel and phase, divided by the mean baseline firing rate across phases (computed as described above). We then plotted the modulation against the baseline firing rate for each channel and phase. To the extent that differences in modulation strength reflected a saturation effect, we expected a negative relationship between modulation strength and baseline firing rate."

Reviewer #2 (Remarks to the Author):

The resubmission generally does a decent job of addressing the previous reviewers' comments. However, it is of concern that the manuscript resubmission removed significant portions of Supplementary Figure 11 that were in the first submission (specifically the sensory projection maps, which in my opinion, showed data that was not just inconclusive, but perhaps even cast doubt on the overall claim of strong alignment between the M1 and S1 fields.). Rather than simply removing that perhaps inconvenient and hard to explain data (namely, that for 2 of 3 subjects, the M1 and S1 fields really did not overlap, nor even have visually similar centroids of spatial activation), that particular strong claim of the manuscript should be reinterpreted. The caveats given for P2 (array age, lack of S1 activation) are understandable, but given the previously presented S1 and M1 maps of the previous Supplementary Figure 11 (which seems to be a much more direct analysis of the shared somatotopy data than the new Figure 5), it is not clear that the conclusion of overlapping fields is as strong as impressed by the manuscript.

We realized, based on the reviewers' comments, that these figures, by showing only dominant digits on the M1 and S1 arrays, provided an incomplete picture of the underlying somatotopic organization, thereby obscuring the point we were trying to get across. Indeed, individual M1 or S1 channels are not restricted to a single digit and the single digit maps do not always reflect the digit gradients. To remedy this, we developed a new visualization of the somatotopic pattern in M1 that explicitly gauges the position of each electrode on a medial to lateral gradient (Supplementary Figure 11). The hue of on each pixel on the heat map indicates the degree to which motor or ICMS-evoked responses on a given channel were dominated by thumb or dominated by little finger, or somewhere in between. This metric more clearly reveals the finger dominance gradients on each array for the motor and sensory responses in M1 and makes these gradients easier to compare by eye.

This visualization is described in the Methods as follows: "To visualize the array-wide somatotopic organization of the motor map, we calculated the Spearman correlation between the motor activation of each electrode and the corresponding digit (thumb = 1, index = 2, ..., pinky = 5). Accordingly, channels that responded preferentially to the lateral digits (ring and little finger) yielded positive correlations; channels that responded preferentially to the medial digits (thumb and index) yielded negative ones. To visualize the array-wide somatotopic organization of the sensory projection map, we calculated the Spearman correlation between ICMS-evoked activation by digit and digit. Accordingly, channels that were most activated by S1 channels with projected fields on the lateral digits yielded positive correlations; channels that were preferentially activated by S1 channels with projected fields on the medial digits yielded negative correlations. The resulting maps revealed preference gradients across the arrays (Supplementary Figure 11)." And the resulting figure is Supplementary Figure 11 in the manuscript.

Reviewers' Comments:

Reviewer #1:

Remarks to the Author:

I would like to thank the authors for the modifications made to the manuscript and the additional analyses carried out in response to my comments. In particular, the new Supplementary Figure 16 clearly shows the absence of an unequivocal relationship between the level of baseline activity and the response to stimulation in the 3 movement phases, either negatively due to a saturation effect, or positively due to a facilitating effect of baseline activity. On the basis of these new results, I now accept the manuscript for publication in nature communication.

Reviewer #2:

Remarks to the Author:

The authors have addressed my final concerns in the addition of the new Supplemental Figure 11 and associated text.